# Prevalence and Associated Factors of Malnutrition and Sarcopenia in a Daycare Facility: A Cross-Sectional Study

**DOI:** 10.3390/healthcare8040576

**Published:** 2020-12-18

**Authors:** Aya Shiota, Naomi Nakayama, Yuki Saito, Tamako Maeda, Yukie Maeda, Kentaro Nakayama

**Affiliations:** 1Social Welfare Corporation Mitsuwa Fukushikai, Himeji 679-2115, Japan; marigold.aya.bd@gmail.com (A.S.); fibonacci_numbers_01123581321@yahoo.co.jp (Y.S.); shinnosuke.of.rikartier@docomo.ne.jp (T.M.); m-yukie@xb3.so-net.ne.jp (Y.M.); kn88@med.shimane-u.ac.jp (K.N.); 2Faculty of Health and Nutrition, The University of Shimane, Izumo 693-8550, Japan; 3Department of Gynecology, School of Medicine, Shimane University, Izumo 693-8501, Japan

**Keywords:** malnutrition, sarcopenia, physical activity, daycare facility, GLIM, AWGS

## Abstract

Malnutrition and sarcopenia are prevalent and growing health issues in older populations. Early detection is important to implement proper interventions. However, little is known about malnutrition and sarcopenia in daycare facilities, the most dominant long-term care service. In this study, the prevalence of and factors associated with malnutrition and sarcopenia in older individuals who commute to community daycare facilities were evaluated. The cross-sectional study included 62 older individuals screened for malnutrition and sarcopenia on their first day in a daycare facility in Japan. Daily physical activity and basal diseases were also evaluated. According to Global Leadership Initiative on Malnutrition (GLIM) criteria, 40.3% (25/62) of patients were malnourished and 59.7% (37/62) were well nourished. The Asian Working Group for Sarcopenia 2019 evaluation found that 12.9% (8/62) of patients showed no sarcopenia, whereas 87.1% (54/62) had sarcopenia. The prevalence of well-nourished sarcopenic individuals was the highest (45.2% (28/62)), followed by malnourished sarcopenia individuals (40.3% (25/62)). All malnourished individuals were sarcopenic and 14.5% (9/62) were well nourished and nonsarcopenic. Daily physical activity was significantly lower among sarcopenic individuals. Subgroups showed no significant difference in comorbidities. The prevalence of malnutrition and sarcopenia was relatively high. Activity-related sarcopenia seemed to precede malnutrition. Early detection of malnutrition and sarcopenia in daycare facilities should be encouraged for early intervention.

## 1. Introduction

Malnutrition is a major cause of adverse health problems in community-dwelling older populations and can result in high mortality, disability, reduced physical function, falls, institutionalization and hospitalization [1,2,3].

The risk of malnutrition increases with age [4] and is mainly caused by inadequate nutritional intake [5]. Older individuals in rehabilitation settings have the highest rate of malnutrition at 80% [6]. Although malnutrition is much lower in community-dwelling older individuals, the overall burden is quite high since older individuals live in their community until an advanced age [6].

The prevalence of malnutrition varies widely depending on the definition applied [7]. The Global Leadership Initiative on Malnutrition (GLIM) criteria provided the first consensus-based universal definition of malnutrition suitable for adults in every healthcare setting [7]. Several studies have reported malnutrition prevalence as evaluated by the GLIM criteria and explored the adverse effects in a community setting [8,9]. Sanchez-Rodriguez et al. reported that 23.4% of community-dwelling older individuals in Belgium were malnourished, according to the GLIM criteria [8]. In addition, malnourished older individuals had a 4.4-fold higher mortality risk [8].

Sarcopenia is characterized by age-related decreases in muscle mass and muscle strength and has received research attention in recent years [10]. Studies have shown that the prevalence of sarcopenia is between 5–29% in older individuals living in the community [11,12,13,14,15,16,17,18], 14–85.4% for those living in long-term care institutions, such as nursing homes, [19,20,21,22], and 10–24.3% for adults in acute hospital care [23,24]. Sarcopenia is an independent risk factor for adverse outcomes, including difficulties in instrumental and basic activities of daily living (ADL) [11,19,20], falls [20], osteoporosis [25], longer length of hospital stay, re-admission [24] and mortality [11]. Since an ICD-10-MC Diagnosis Code was issued for sarcopenia in 2017, many international academic societies have focused on identifying older individuals with sarcopenia for intervention. To address ethnic diversity in body composition, the Asian Working Group for Sarcopenia (AWGS) proposed its diagnostic criteria for people in Asia in 2014 (Asian Working Group for Sarcopenia criteria, 2014, AWGS 2014) and revised it in 2019 (AWGS 2019) [26,27]. Of particular interest in the AWGS 2019 is the proposal for separate algorithms for primary healthcare and hospital settings to enable earlier identification of people with sarcopenia. The primary healthcare algorithm for identifying individuals who require proper intervention for sarcopenia begins with the assessment of either calf circumference (CC) (<34 cm in men, <33 cm in women), followed by a SARC-F questionnaire for diagnosing sarcopenia (≥4), or SARC-CaIF sarcopenia questionnaire (≥11), and then an assessment of muscle strength and physical function through handgrip strength and the five times sit-to-stand test [27]. All the steps can be conducted without the special medical devices, such as bioelectrical impedance analysis (BIA) and dual-energy X-ray absorptiometry (DEXA), that are unavailable in primary healthcare and community settings [27].

Both malnutrition and sarcopenia are well recognized as serious health conditions related to adverse consequences and worse outcomes in hospital care settings [28]. Compared to hospital care settings, diagnosis is delayed in primary healthcare and community settings, even though both malnutrition and sarcopenia are growing health problems in the aging community. There is less general recognition and awareness of sarcopenia than malnutrition because it is a newer concept. This is even more evident in community settings than in hospitals since a screening tool for sarcopenia, which can be incorporated into community settings, has only recently been created [27]. In Japan, an additional fee has recently been introduced for malnutrition screening in long-term care facilities, approximately 15 years after the fee was introduced in hospitals [29].

This study was conducted at a daycare facility in Japan. About 12% of the aged population in Japan uses one or more long-term care service [30]. In Japan, there are over 1.13 million clients of daycare facilities utilizing several different services; daycare services are the most utilized among community-living older individuals who are not fully independent [30]. In general, daycare service providers aim to offer care services that not only support clients’ health and daily living needs but also improve their continued independence [31].

Thus, malnutrition and sarcopenia should be identified and addressed in daycare facilities, as both are related to functional decline and adverse outcomes [8,9,11,19,20,25]. However, there have been few reports in this field of research from daycare facilities. Therefore, this study aimed to determine the prevalence and associated factors of malnutrition and sarcopenia in daycare facilities in order to increase recognition and awareness of these factors.

## 2. Materials and Methods 

### 2.1. Population

A cross-sectional study was conducted at a single daycare facility in a local community in Japan. A total of 62 older individuals aged >65 years who started to commute to this facility from their homes between January and June 2019 were included. Diagnosis of malnutrition and sarcopenia and assessment of covariates were carried out on their first day in the facility.

Written informed consent was obtained from all the patients for the publication of this paper. The protocol and study design were approved by the ethics committee of the University of Shimane (IRB: 321). 

### 2.2. Diagnosis of Malnutrition 

Malnutrition was screened and diagnosed on the first day at the daycare facility according to the Mini Nutritional Assessment—Short Form (MNA-SF) screening method and the GLIM criteria for diagnosis [32].

Individuals who were categorized as malnourished or at risk of being malnourished using the MNA-SF were then examined using the GLIM criteria to diagnose malnutrition. 

For phenotypic criteria, the cut-off value for low body mass index (BMI) was incorporated using the reference for Asians, <18.5 if <70 years, or <20 if >70 years. Reduced muscle mass was measured using calf circumference (CC). The cut-off value was incorporated from sarcopenia diagnostic criteria for Asian populations (AWGS2019), <34 cm for men and <33 cm for women, as measured with dual-energy absorptiometry (DXA) or corresponding standards using other body composition methods, such as bioelectrical impedance analysis (BIA), CT and MRI were not available in this research setting. 

For etiologic criteria, inflammation was measured using the participants’ comorbidities that caused chronic disease-related inflammation, as people suffering from acute disease and injury did not appear at this daycare facility. Most chronic organ diseases, such as congestive heart failure, chronic obstructive pulmonary disease, rheumatoid arthritis, chronic kidney or liver disease, as well as cancer, are associated with chronic or recurrent inflammation of a mild to a moderate degree [30]. Weight loss in phenotypic criteria and reduced food intake or assimilation in etiologic criteria were measured by interview. 

### 2.3. Diagnosis of Sarcopenia

To diagnose sarcopenia, we applied the latest AWGS2019 criteria published by the AWGS [26] and used the algorithms for primary healthcare. Sarcopenia was assessed on the participant’s first day at the daycare facility. Low muscle mass was defined as CC <34 cm for males and <33 cm for females. Individuals who were categorized as having low muscle mass and/or were positive for SARC-F (≥4) or SARC-CaIF (≥11) were subjected to a follow-up examination of muscle strength and physical function [27,33,34]. Reduced muscle strength was defined as handgrip strength <28 kg for male and <18 kg for female participants. Physical function was assessed using the five times sit-to-stand test. Participants were considered to have a low physical function when they took longer than 12 s to complete the test or could not complete it. When decreased muscle strength and/or decreased physical function were confirmed, participants were diagnosed with sarcopenia. Of these diagnosed participants, those with decreased physical function were categorized as having severe sarcopenia. 

### 2.4. Assessment of Covariates

In this study, several covariates were also collected. Comorbid conditions were scored using the Charlson Index as described by Charlson et al. [35]. The Charlson Index considers 19 different comorbidities, each classified with a score of 1–6 points, based on the adjusted relative risk of 1-year mortality. Each participant’s comorbidities resulted in a final score that was used to calculate the probability of survival. A final score of 0 points indicated a one-year mortality risk of 12%, 26% for 1–2 points, 52% for 3–4 points and 85% for ≥5 points. The Charlson Index score was calculated based on the medical history at admission.

The Japan Public Health Center-based Prospective Study—Physical Activity Questionnaire (JPHC-PAQ) was used to evaluate total daily physical activity (total PA) in this study. The JPHC-PAQ is a common validation tool that has been used in Japanese large-sized cohort studies. The validity and reliability of JPHC-PAQ for estimating total PA has been reported elsewhere [36,37,38,39,40]. The JPHC-PAQ was calculated based on the participants’ interviews on their first day at the daycare facility.

### 2.5. Statistical Analyses

IBM SPSS Statistics for Windows, version 26 (IBM Corp., Armonk, NY, USA), was used for analysis. Continuous variables were reported as means (standard deviation (SD)) and categorical variables as numbers (%). A sample estimate univariate analysis was performed to examine factors associated with sarcopenia and malnutrition. Statistical significance was set at *p* < 0.05.

## 3. Results

The characteristics of the study participants are summarized in Table 1. A total of 62 older individuals (74.2% female) aged 71–100 years were recruited. The mean age was 86.6 years. According to the long-term care insurance system for older individuals, older individuals who are not fully independent are categorized into seven nursing care levels, from grade 1 to grade 7, based on the assessment of care requirements. Grade 1 is the least severe and grade 7 is the most severe in terms of their functional ability level. Therefore, elderly individuals in grade 7 are fully dependent or bed-ridden, while those in grade 1 need only partial support for activities such as grocery shopping and laundry. The most prevalent grade in daycare facilities is grade 4; elderly individuals in this grade require partial care for basic daily living activities such as eating, toileting, bathing and changing clothes. Those in grade 5, the second most prevalent in daycare facilities, need more care for basic daily living than those with grade 4. Most participants (32.4%) were categorized in grade 4. 

### 3.1. Population and Diagnosis of Malnutrition and Sarcopenia

Of the 62 older individuals screened by MNA-SF, 26 (41.9%) were well nourished, while 36 (58.1%) were at risk of being malnourished or were malnourished. Of the 36 individuals who were either at risk of being malnourished or were malnourished, 25 (40.3%) were diagnosed with malnutrition according to the GLIM criteria.

Only 8 (12.9%) participants were not sarcopenic according to the AWGS2019 primary healthcare criteria. Of the remaining 54 individuals, 39 (62.9%) were categorized as having severe sarcopenia with decreased muscle strength and physical function, while 15 (24.2%) showed only decreased muscle strength.

### 3.2. Evaluation of Comorbid Conditions and Daily Physical Activity

The mean Charlson Comorbidity Index was 1.1 and its range was from 0 to 6. Specifically, 21 (33.9%) individuals were categorized as low, 36 (58.1%) as medium, 4 (6.5%) as high and 1 (1.6%) as very high risk. The mean JPHC-PAQ score was 28.1 and its range was from 15.25 to 36.50. 

### 3.3. Factors Associated with Malnutrition and Sarcopenia

Table 2 and Table 3 show the factors significantly associated with malnutrition and sarcopenia. In addition to nutritional and body composition factors, such as BMI, MNA-SF, CC and SARC-CaIF, univariate analysis showed that the participants’ age was associated with malnutrition (*p* < 0.05). In addition to the factors related to muscle volume, muscle strength and physical function, others, such as those related to nursing care level, CC, handgrip strength, the five times sit-to-stand test, the Timed Up and Go test and SARC, as well as nutrition-related factors and physical activity levels, such as those measured by BMI, MNA-SF and JPHC, were associated with sarcopenia in the univariate analysis (*p* < 0.05). 

The comorbid conditions evaluated by the Charlson Comorbidity Index did not show any relationship with either malnutrition or sarcopenia.

### 3.4. Factors Associated with Nutrition Status and Sarcopenia

Figure 1 shows study participants categorized according to their nutrition and sarcopenia status. Most individuals were classified as well-nourished but sarcopenic (54.2%), followed by individuals who were malnourished and sarcopenic (40.3%). All malnourished participants were sarcopenic and only 9 (14.5%) individuals were neither malnourished nor sarcopenic.

### 3.5. MNA-SF Score, Comorbidity, JPHC-PAQ Association with Nutritional and Sarcopenia Status

The association between the MNA-SF score, comorbidity, JPHC-PAQ score and nutritional and sarcopenia status was analyzed using the box-and-whisker plot method. The lower, middle and upper hinge of the box correspond to the 25th, 50th and 75th percentiles, respectively. The rhombus indicates the mean. The MNA-SF score was significantly higher in individuals who were not malnourished and sarcopenic among all the three subgroups (13.11 ± 1.0 vs. 11.50 ± 1.4, *p* < 0.001; 13.11 ± 1.0 vs. 9.88 ± 1.6, *p* < 0.001). It was also significantly higher in well-nourished individuals who were sarcopenic compared to the malnourished individuals who were sarcopenic (11.50 ± 1.4 vs. 9.88 ± 1.6, *p* < 0.05) (Figure 2). The JPHC-PAQ score was significantly higher in those who were not malnourished and sarcopenic (33.61 ± 5.7 vs. 27.10 ± 5.5, *p* < 0.001; 33.61 ± 5.7 vs. 27.25 ± 5.7, *p* < 0.05). There was no significant difference between well-nourished individuals who were sarcopenic and malnourished individuals who were sarcopenic (27.10 ± 5.5 vs. 27.25 ± 5.7, *p* > 0.05) (Figure 3). Comorbidity status had no relationship with any subgroup of participants (*p* > 0.05).

The MNA-SF score was significantly higher in those who were neither malnourished nor sarcopenic than in the other subgroups (13.11 ± 1.0 vs. 11.50 ± 1.4, *p* < 0.001; 13.11 ± 1.0 vs. 9.88 ± 1.6, *p* < 0.001). It was also significantly higher in well-nourished individuals who were sarcopenic than in malnourished individuals who were sarcopenic (11.50 ± 1.4 vs. 9.88 ± 1.6, *p* < 0.05).

The JPHC-PAQ score was significantly higher in those who were neither malnourished nor sarcopenic than in the other subgroups (33.61 ± 5.7 vs. 27.10 ± 5.5, *p* < 0.001; 33.61 ± 5.7 vs. 27.25 ± 5.7, *p* < 0.05). There was no significant difference between well-nourished individuals who were sarcopenic and malnourished individuals who were sarcopenic (27.10 ± 5.5 vs. 27.25 ± 5.7, *p* > 0.05).

## 4. Discussion

To the best of our knowledge, this is the first study to investigate the prevalence of and factors associated with malnutrition and sarcopenia in a daycare facility. In this study, we incorporated the latest diagnostic tools for assessing both malnutrition (GLIM criteria) and sarcopenia (AWGS 2019) in a community setting and demonstrated that the prevalences of malnutrition (40.3%) and sarcopenia (87.1%) in daycare facilities were relatively high compared to the results in previous reports. 

The GLIM criteria were recently published by the Global Leadership Initiative on Malnutrition to support a global consensus on malnutrition diagnosis in different clinical settings [5], since the prevalence of malnutrition varies based on the choice of diagnostic tools [41]. It has been reported that 23.4% of community-dwelling older individuals meet the GLIM criteria for malnutrition, which was associated with 4.4-fold higher mortality during the four-year follow-up in European countries [8]. Another study from Spain reported that 12.6% of community-dwelling older adults were malnourished according to the GLIM criteria and noted a further association with increased medical cost [41]. In the hospital setting, the prevalence of GLIM-diagnosed malnutrition was higher, 46% in hospitalized older patients [41]. In this study, the prevalence of malnutrition in daycare facilities was larger than that recorded among community-dwelling older adults but lower than that among hospitalized older adults. These findings indicate that community-dwelling people who use daycare services are prone to malnutrition. It is plausible that older people start to use daycare services when their independence starts to decline. It has also been reported that the prevalence of malnutrition increases with dependency level in older people [42]. We also discovered that the prevalence of sarcopenia was quite high (87.1%) among older individuals who commute to daycare facilities in the community. Just like malnutrition, the prevalence of sarcopenia varies depending on the age group and clinical setting. Diagnostic tools and ethnicity also affect prevalence. In previous studies, the prevalence of sarcopenia was 1–29% in community-dwelling older adults [11,12,13,14,15,16,17,18], 14–85.4% in nursing homes [19,20,21,22] and 10–24.3% for adults in hospitals [23,24]. In Japan, the most widely utilized criteria for determining sarcopenia are based on the AWGS published in 2014 [26]. The AWGS 2014 criteria are consistent with those of the European Working Group on Sarcopenia in Older People (EWGSOP); however, the cut-off values have been revised according to data from regional cohort studies among Asian populations [26]. AWGS 2019, the revised version of AWGS 2014, provides separate algorithms to facilitate earlier identification of people with sarcopenia in both primary healthcare and the community. We incorporated the AWGS 2019 criteria in this study and discovered that the prevalence of sarcopenia in daycare facilities was almost the same as the highest prevalence among the reports from nursing homes (85.4%) [22]. As mentioned, this is plausible because older individuals in the community start to commute to daycare services as they become older and more dependent. From these results, we can conclude that older individuals in the community, who utilize daycare facilities, are at high risk of malnutrition and sarcopenia. 

It has been reported that malnutrition is strongly associated with sarcopenia in community-dwelling older individuals [28]. This is consistent with our results, which revealed that nutritional status was significantly associated with the presence of sarcopenia. Anorexia, induced by several factors that occur during aging, is a common condition in older individuals [5]. Anorexia, combined with other aging-related problems, such as denture, cognitive and socioeconomic problems, results in malnutrition. Malnutrition in older individuals is usually linked to inadequate protein intake, the building blocks for muscle metabolism [43,44]. This causes muscle wasting and the development of sarcopenia. Indeed, malnutrition is the main cause of sarcopenia among community-dwelling older individuals and precedes sarcopenia [43,44]. Among sarcopenic individuals in our study, those who were not malnourished were predominant relative to those who were malnourished. This indicates that other causal factors are dominant in the development of sarcopenia among older people who use daycare services. In this study, we evaluated comorbid conditions and total daily physical activity using a well-validated scoring tool and showed that daily physical activity was significantly associated with the presence of sarcopenia.

Sarcopenia is characterized by a progressive and generalized loss of skeletal muscle mass and strength and is categorized as age-, activity-, nutrition- and disease-related [45]. In general, activity-related sarcopenia arises from prolonged bed rest, deconditioning or a sedentary lifestyle; nutrition-related sarcopenia is caused by inadequate energy and/or protein intake; and disease-related sarcopenia results from acute illness, injury and/or cachexia induced by chronic illness [45]. Of these causal factors, age is unavoidable. Therefore, it is important to avoid exposure to other factors to prevent and improve sarcopenia in older individuals. Age-, activity-, nutrition- and disease-related sarcopenia frequently overlap in hospitalized patients. Hospitalized older patients can easily develop iatrogenic sarcopenia from unnecessary bed rest and inappropriate nutritional care management in addition to the disease-related sarcopenia that requires in-hospital care [45,46]. In this study, all individuals were quite old (mean age, 86.6 years), suggesting that age likely plays an essential role in sarcopenia development. In addition, age and activity-related sarcopenia overlap more frequently than age and nutrition-related sarcopenia [45]. In fact, the prevalence of nonmalnourished sarcopenic individuals was the highest among all our subgroups and daily physical activity level was significantly associated with sarcopenia status. For older individuals in this subgroup, activity-related sarcopenia seemed to precede malnutrition, which then caused nutrition-related sarcopenia.

The term “sarcopenic dysphasia” was recently coined to describe impairments in swallowing due to a generalized loss of skeletal muscle mass and strength [47,48]. The first position paper on sarcopenic dysphasia was published in January 2019 and described the diagnostic criteria (age, general muscle mass, handgrip strength, gait speed, impaired swallowing and absence of cause of dysphasia) and general therapeutic management, including resistance training of the swallowing muscles and nutritional care [49]. It has also been reported that, in older individuals, swallowing is significantly impaired with sarcopenia before clinical dysplasia symptoms become clear [50]. Sarcopenic older individuals have reduced tongue pressure, reduced swallowing speed and increased hyoid bone movement even though they do not show clinical dysphasia [50]. Impaired swallowing and mastication ability may cause an unbalanced diet and dietary limitations, resulting in malnutrition, even before dysphasia becomes clinically evident. Reduced muscle mass also induces a reduction in basal energy metabolism, causing fatigability and sedentary lifestyle, thereby inducing appetite loss. In this regard, malnutrition is both a causal factor of sarcopenia and a consequence of it.

Currently, there are over 36 million people above 65 years of age living in Japan, which constitutes around 28.7% of the entire Japanese population [51]. In 2019, the average life expectancy was estimated to be 81.41 years for men and 87.45 years for women and continues to increase [27]. Predicted changes in the demographic structure will result in several challenges in planning and shaping the social policy related to older adults. The aging population will require more organized systems to support their daily activities and improve their general health. Therefore, the Japanese government has undertaken actions to meet the broad and complex needs of older adults. 

Daycare is one of the long-term care (LTC) services provided at daycare facilities. To receive LTC services, elderly individuals are required to obtain certification of care needs from the local government. Certification has seven nursing care levels, from grade 1 to grade 7, depending on an individual’s functional ability. Daycare facilities provide day services to elderly individuals who commute from their home. Therefore, daycare facilities have a lighter client dependency level than nursing homes. While daycare services are available for elderly individuals from any nursing care grade, nursing home care is only accessible to those requiring at least grade 5 nursing care. Clients can choose a particular daycare facility and the frequency of commuting depends on their LTC coverage and needs. Daycare facilities generally operate from morning to evening and offer several services to the clients. These services are heterogeneous and range from daily activities, such as meal and snack provision and bathing, to interventional activities, such as exercising, games, hobbies and rehabilitation. About 12% of the aged population in Japan used one or more LTC service in April 2018 [52]. National LTC insurance covers LTC services and people who are not fully independent (over 97% of those reported in the study were 65 years or over) can use the LTC services under this insurance fee coverage [40]. In 2020, the number of daycare service clients was 1.13 million and daycare was the most popular service among the population using LTC services [52]. One international review demonstrated that daycare service providers had the following general aims: (i) providing social and preventive services, (ii) supporting clients’ continued independence and (iii) supporting attendees’ health and daily living needs [31]. 

To achieve these aims, early detection of malnutrition and sarcopenia should be encouraged in daycare facilities since the necessary diagnostic tools are available and malnutrition and sarcopenia are associated with several adverse health problems. This could lead to proper intervention and prevention in daycare facilities. 

This study had some limitations. First, we used the AWGS 2019 primary healthcare algorithm to diagnose sarcopenia. Since it does not require the precise measurement of muscle mass by BIA or DEXA, the accuracy of diagnosis may be inferior to that of a hospital. Second, we did not assess swallowing ability for comparison between subgroups. Third, this study was conducted in a single facility and included older individuals within that location only. A follow-up study, including expanded target facilities, is warranted.

## 5. Conclusions

We discovered that the prevalences of malnutrition and sarcopenia were relatively high (40.3% and 87.1%, respectively) among community-dwelling older individuals in daycare facilities. Under-recognition of malnutrition and sarcopenia is problematic because they are both associated with adverse healthcare problems [1,2,3,11,19,20,24,25]. The prevalence of nonmalnourished sarcopenic individuals was higher than that of malnourished sarcopenic individuals. These are inseparable conditions that induce each other. Early detection of both malnutrition and sarcopenia is necessary for early intervention in daycare facilities.

## Figures and Tables

**Figure 1 healthcare-08-00576-f001:**
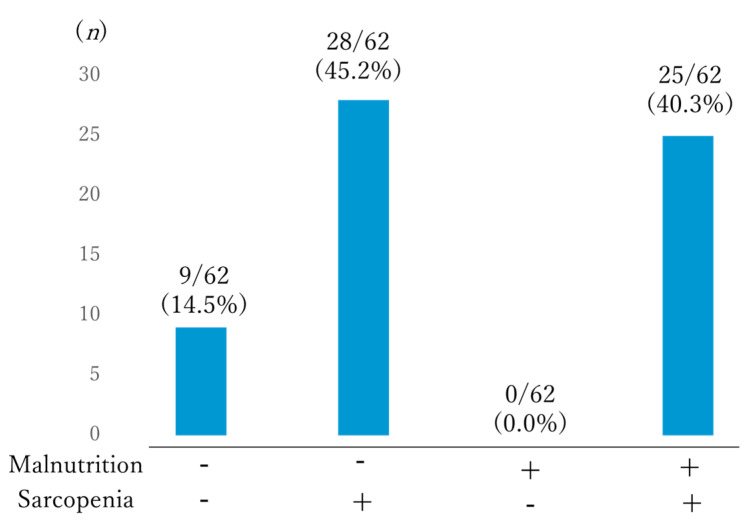
Subgroups categorized according to nutritional and sarcopenia status.

**Figure 2 healthcare-08-00576-f002:**
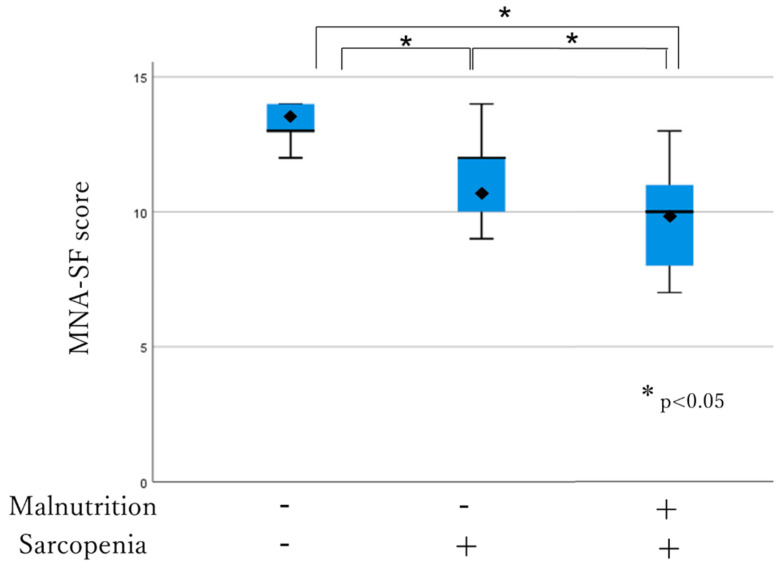
Subgroup association with MNA-SF.

**Figure 3 healthcare-08-00576-f003:**
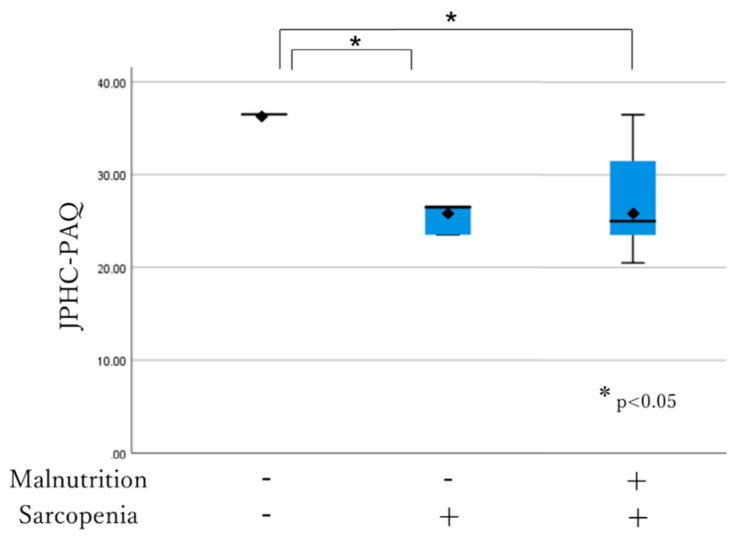
Subgroup association with JPHC-PACQ score.

**Table 1 healthcare-08-00576-t001:** Characteristics of the study population.

Characteristics		Total (*n* = 62)
Age, years, mean (SD)		86.6 (7.0)
Female sex, *n* (%)		46 (74.2)
Nursing care level, *n* (%)	grade1	4 (6.5)
	grade2	5 (8.1)
	grade3	10 (16.1)
	grade4	20 (32.3)
	grade5	14 (22.6)
	grade6	9 (14.5)
	grade7	0 (0.0)
BMI, kg/m^2^, mean (SD)		23.6 (4.1)
BMI category, *n* (%)	underweight (<22)	23 (37.1)
	normal weight (22–25)	15 (24.2)
	overweight (>25)	24 (38.7)
MNA-SF, *n* (%)	malnourished	2 (3.2)
	at risk of malnutrition	34 (54.8)
	wellnourished	26 (41.9)
GLIM, *n* (%)	malnourished	37 (59.7)
	wellnourished	25 (40.3)
Sarcopenia, *n* (%)	no-sarcopenia	8 (12.9)
	sarcopenia	15 (24.2)
	severe sarcopenia	39 (62.9)
JPHC-PAQ score, mean (SD)		28.1 (6.0)
Charlson Comorbidity Index, mean (SD)		1.1 (1.1)

SD: standard deviation; BMI: Body Mass Index; MNA-SF: Mini Nutritional Assessment—Short Form; GLIM: Global Leadership Initiative on Malnutrition; JPHC-PAQ: Japan Public Health Center-based Prospective Study—Physical Activity Questionnaire.

**Table 2 healthcare-08-00576-t002:** Factors associated with malnutrition.

Mean (±SD)	GLIM Criteria		*p*-Value
	Well Nourished (*n* = 37)	Malnourished (*n* = 25)	
Age years *	84.92 (7.1)	89.24 (6.2)	0.0017
Nursing care level	2.95 (1.2)	3.08 (1.5)	0.71
BMI, kg/m^2^ *	25.34 (3.6)	21.13 (3.6)	<0.001
MNA-SF *	11.89 (1.5)	9.88 (3.6)	<0.001
Charlson Comorbidity Index	0.94 (1.1)	1.28 (1.1)	0.259
JPHC-PAQ score	28.79 (6.1)	27.25 (5.7)	0.332
CC (cm) *	33.65 (3.2)	29.6 (2.1)	<0.001
Hand grip strength (kg)	14.25 (5.3)	12.8 (6.4)	0.343
Five time chairstand up test (s)	15.91 (7.0)	15.37 (5.5)	0.797
SARC-F	3.37 (3.0)	4.68 (3.5)	0.129
SARC-CaIF *	7.97 (6.1)	14.68 (3.5)	<0.001
TUG (s)	16.03 (7.0)	17.27 (5.2)	0.574

SD: standard deviation; BMI: Body Mass Index; MNA-SF: Mini Nutritional Assessment—Short Form; GLIM: Global Leadership Initiative on Malnutrition; JPHC-PAQ: Japan Public Health Center-based Prospective Study—Physical Activity Questionnaire; CC: Calf Circumference; SARC-F: Strength, Assistance with walking, Rise from a chair, Climb stairs, and Falls; SARC-CaIF: SARC-F and Calf Circumference; TUG: Timed Up and Go test; * *p* < 0.05.

**Table 3 healthcare-08-00576-t003:** Factors associated with sarcopenia.

Mean(±SD)	AWGS2019		*p*-Value
	Non-Sarcopenia (*n* = 9)	Sarcopenia (*n* = 53)	
Age years	84.11 (6.3)	87.09 (7.1)	0.24
Nursing care level *	2.33 (0.7)	3.11 (1.4)	0.018
BMI, kg/m^2^ *	28.01 (2.8)	22.90 (3.9)	<0.001
MNA-SF *	13.11 (1.0)	10.73 (1.7)	<0.001
Charlson Comorbidity Index	0.66 (0.7)	1.15 (1.1)	0.259
JPHC-PAQ score *	33.61 (5.7)	27.22 (5.5)	0.01
CC (cm) *	36.22 (2.9)	31.31 (3.0)	<0.001
Hand grip strength (kg) *	18.87 (5.5)	12.78 (5.3)	0.003
Five time chairstand up test (s) *	12.21 (2.2)	16.66 (6.9)	0.003
SARC-F *	0.55 (0.8)	4.47 (3.2)	<0.001
SARC-CaIF *	0.55(0.8)	12.39 (4.8)	<0.001
TUG (s) *	12.08 (2.2)	17.65 (6.7)	0.021

SD: standard deviation; AWGS: Asian Working Group for Sarcopenia; CC: Calf Circumference; SARC-F: Strength, Assistance with walking, Rise from a chair, Climb stairs, and Falls; SARC-CaIF: SARC-F and Calf Circumference; TUG: Timed Up and Go test; BMI: Body Mass Index; MNA-SF: Mini Nutritional Assessment—Short Form; JPHC-PAQ: Japan Public Health Center-based Prospective Study—Physical Activity Questionnaire; * *p* < 0.05.

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
