# Peer review of "Prevalence and Associated Factors of Malnutrition and Sarcopenia in a Daycare Facility: A Cross-Sectional Study"

_healthcare, 2020, doi:10.3390/healthcare8040576_

Round 1

Reviewer 1 Report

This manuscript is well-written and provide modelately useful information regarding the related factors to malnutrition and sarcopenia in elderly daycare facility. One concern is that MNA-SF screening method does covers not only causes (reduced food intake, swallowing difficulty) of  malnutrition but also results (weight loss, low physical activity, reduced CC) of malnutrition. I suggest that author can reanalyze MNA-SF scores regarding cause and result sections of malnutrition to present clear relationship between factors and malnutrition. In addition, the quality of meal should be matter for malnutrition. Therefore, author should show that the meal information whether all participants were provided same meals in the daycare or had different meals individually. 

Minor 1: double punctation mark at line 32. Need proofreading.

Minor 2: Figure 1 and 2 legend is duplicated in the result section. Figure legends provide the information about the study design and results without reading result section.  

Author Response

We wish to express our appreciation to the reviewer for his or her insightful comments, which have helped us significantly improve our manuscript.

We have provided point-by-point responses to your comments.

Major comment

Answer:

Thank you very much for your suggestion. According to the instruction of GLIM criteria, those who are categorized in malnutrition or risk of malnutrition should be evaluated by GLIM criteria for final diagnosis of malnutrition. Therefore, as we mentioned in manuscript, we evaluated the prevalence of malnutrition by using GLIM criteria for final diagnosis. GLIM criteria include causes and results as phenotypic criteria and etiologic criteria and malnutrition diagnosis requires at least one positive item form both criteria. Therefore, individuals who are diagnosed as malnutrition fulfilled both criteria in this study. However, your suggestion is valuable, and we would like to analyze it in future research. 

As I added in manuscript on page 11, lines 326-337, unlike nursing home or assisted care home, individuals commute daycare from their home. Lunch and snack are basically provided at most daycare facilities in Japan and there is no nationwide standardized menu for meal provision at daycare facilities. Malnutrition and sarcopenia were investigated at the first day of daycare services in this research. Therefore, I did not mention the detail of meal information of this daycare facility in our manuscript because it does not affect the result. However, after this research was conducted, this facility staffs started to provide personalized meal menu to malnourished and/or sarcopenic individuals to improve their condition.   

Minor comments

Minor 1:

Answer:

Line 32: I deleted one punctuation mark as indicated: intervention.

Minor 2:

Answer 2:

I deleted the GLIM part in line 199 as table 3 does not contain the abbreviation, GLIM.

Reviewer 2 Report

Journal: Healthcare, December 2020

Article: Prevalence and Associated Factors of Malnutrition 3 and Sarcopenia in a Daycare Facility: A Cross4 Sectional Study

Manuscript ID: healthcare-1043978

REVIEW

Overall thoughts:

A well-written study of practical importance to elder public health. There are more than a few areas where grammar, punctuation, and word choice are not ideal – I strongly recommend a close review with a grammar/language expert.

I believe it would be helpful for a readership audience who is unaware, what the characteristics of a daycare community are - how are they different from a nursing care unit, how are they different from assisted living, what percentage of patrons commute vs live on the premises, et cetera.

What typically happens at these locations daily? … tell the reader what is going on. What functions does the daycare facility provide? Food, exercise, social interaction, et cetera. Are there information sessions with patrons regarding muscle maintenance, nutrition recommendations et cetera? … perhaps you can make these recommendations for daycare centers in your discussion.

Is food provided and how much? What is the ratio of macro nutrients and calorie count.

What is the ‘functional ability’ of the participants in this study? What other physical/medical conditions to they express? What is their relationship with the daycare center? What is their level of dependence? … I believe these are all very important contextual variables to state.

BMI is tremendously flawed. With such a small participant size, body fat percentage could be obtained. BMI only has usefulness when measuring a much larger sample. Consider using a different metric to measure participants’ fat/fat-free mass ratios [I understand that access to DEXA or BIA was not available, but I don’t see how the results are verifiable by BMI, alone; this isn’t just a limitation, it’s a methodological deal-breaker]

There appears to be more information describing daycare centers in the discussion that in the introduction – I would reverse that. Same goes for detail describing malnourishment and sarcopenia.

Specific comments:

Line 39. Comma after ‘populations’.

Line 44. ‘till’ should be ‘until’.

Line 45. Here is where indicating a few definitions of malnourishment would be very useful. And CRUCIAL to the context of the present study, indicate which definition the researchers used.

Line 76. “…awareness of sarcopenia than of malnutrition since it is a newer concept.” Consider revising to: …awareness of sarcopenia than of malnutrition, because it is a newer concept.

Line 80. Consider rephrasing or adding detail for clarity.

Line 108. Indicated what “CC” is.

Line 110. Indicated what the abbreviations stand for – I know it is taken for granted that our field understands them, but APA standards indicate writing out the full names at first use.

Author Response

We wish to express our appreciation to the reviewer for his or her insightful comments, which have helped us significantly improve our manuscript.

We have provided point-by-point responses to the reviewer’s comments.

Reviewer’s overall thoughts

Response:

Thank you very much for your comments. We added the following underlined sentences to describe clients’ functional ability on page 4, lines 153-159 in the revised manuscript.

Grade 1 is the least severe, and grade 7 is the most severe according to their functional ability level. Therefore, elderly individuals with grade 7 are fully dependent or bed-ridden, while those with grade 1 need only partial support for activities such as grocery shopping and laundry. The most prevalent grade in daycare facilities is grade 4; elderly individuals in this grade require partial care for basic daily living activities such as eating, toileting, bathing, and changing clothes. Those with grade 5, second most prevalent in daycare facilities, need more care for basic daily living than those with grade 4.

We also added the following underlined sentences to define daycare facilities on page 11, lines 326-337.

To receive LTC services, elderly individuals are required to obtain certification of care needs from the local government. Certification has seven nursing care levels from grade 1 to grade 7, depending on an individual’s functional ability. Daycare facilities provide day services to elderly individuals who commute from their homes. Therefore, daycare facilities have a lighter client dependency level than nursing homes. While daycare services are available for elderly individuals with any nursing care grades, nursing home care is accessible to those with at least grade 5 of nursing care. Clients can choose a particular daycare facility, and the frequency of commuting depends on their LTC coverage and needs. Daycare facilities generally operate from morning to evening and offer several services to the clients. These services are heterogeneous and range from daily activities such as meal and snack provision and bathing to interventional activities such as exercising, games, hobby, and rehabilitation.

As you mentioned, BMI is not an appropriate indicator of malnutrition and sarcopenia. This is because BMI does not reveal the body composition, and sarcopenic obesity is quite common in western countries compared to Asian countries. In this regard, we incorporated GLIM criteria and AWGS 2019 to diagnose malnutrition and sarcopenia, respectively. GLIM criteria constitute the globally standardized malnutrition diagnostic tool, and AWGS 2019 is the diagnostic tool invented specifically for Asian older individuals. BMI is one of the phenotypic criterion of GLIM criteria besides body weight loss and reduced muscle mass. By using GLIM criteria, diagnosis of malnutrition is made through comprehensive evaluation of these phenotypic criteria combined with etiologic criterion described in the methods section on page 3, lines 110-116 in the revised manuscript.

Specific comments

Answer:

I corrected all phrases and sentences described below according to your specific comments in line 37, 42, 74, 78, 106, and 109.

Line 37: populations to populations,

Line 42: till to until

Line 74: “…awareness of sarcopenia than of malnutrition since it is a newer concept.” to: …awareness of sarcopenia than of malnutrition because it is a newer concept.

Line 78: rephrased as approximately 15 years after the fee was effected in hospitals

Line 106: I corrected as calf circumference (CC).

Line 109: I corrected as bioelectrical impedance analysis (BIA)

Round 2

Reviewer 1 Report

Thank you for answers to clarify. Since the title says the malnutrition and sarcopenia is tested elderly of a daycare facility, readers could think that the service of daycare may affect to malnutrition and sarcopenia of elderly. Please emphasize that malnutrition and sarcopenia test was conducted at the first day of daycare, which is related to health status of elderly who need daycare service rather than quality of care of a daycare facility.

Author Response

Answer

We wish to express our appreciation to the reviewer for his or her insightful comments, which have helped us significantly improve our manuscript. 

 According to your suggestion, I made below two modifications in the manuscript to emphasize that malnutrition and sarcopenia test was conducted at the first day of daycare.

1) In line 20, abstract part, I inserted “on their first day” into below sentence.

This cross-sectional study included 62 older individuals screened for malnutrition and sarcopenia on their first day in a daycare facility in Japan.

2) In line 94, material and method part, I added below sentences.

Diagnosis of malnutrition and sarcopenia and assessment of covariates were carried out on their first day in the facility.

Reviewer 2 Report

Thank you, authors, for considering my comments, feedback, and suggestions. I feel adequate attention was given to my most immediate concerns, therefore, I believe your manuscript is well-written, scientifically sounds, and of meaningful relevance to the healthcare community. Thank you for your contribution.

Author Response

Dear. Reviewer

We wish to express our appreciation to the reviewer for his or her insightful comments.

Thank you very much for your generous comment and we are grateful to contribute to this journal.
